# Rhythm profiling using COFE reveals multi-omic circadian rhythms in human cancers in vivo

**Bharath Ananthasubramaniam**[1]*, **Ramji Venkataramanan**[2]

**1** Institute for Theoretical Biology, Humboldt Universität zu Berlin, Berlin, Germany, **2** Department of Engineering, University of Cambridge, Cambridge, United Kingdom

* bharath.ananthasubramaniam@hu-berlin.de

**Data availability statement:** All data and code underlying the findings of this study are fully

## Abstract

The study of ubiquitous circadian rhythms in human physiology requires regular measurements across time. Repeated sampling of the different internal tissues that house circadian clocks is both practically and ethically infeasible. Here, we present a novel unsupervised machine learning approach (COFE) that can use single high-throughput omics samples (without time labels) from individuals to reconstruct circadian rhythms across cohorts. COFE can simultaneously assign time labels to samples and identify rhythmic data features used for temporal reconstruction, while also detecting invalid orderings. With COFE, we discovered widespread de novo circadian gene expression rhythms in 11 different human adenocarcinomas using data from The Cancer Genome Atlas (TCGA) database. The arrangement of peak times of core clock gene expression was conserved across cancers and resembled a healthy functional clock except for the mistiming of a few key genes. Moreover, rhythms in the transcriptome were strongly associated with the cancer-relevant proteome. The rhythmic genes and proteins common to all cancers were involved in metabolism and the cell cycle. Although these rhythms were synchronized with the cell cycle in many cancers, they were uncoupled with clocks in healthy matched tissue. The targets of most of FDA-approved and potential anti-cancer drugs were rhythmic in tumor tissue with different amplitudes and peak times. These findings emphasize the utility of considering "time" in cancer therapy, and suggest a focus on clocks in healthy tissue rather than free-running clocks in cancer tissue. Our approach thus creates new opportunities to repurpose data without time labels to study circadian rhythms.

## Introduction

Near 24h biological rhythms, called circadian rhythms, are essential to healthy human physiology and their disruption is associated with several pathologies [1]. These rhythms are orchestrated by a symphony of biological clocks in most tissues that drive tissue-specific temporal programs [2]. Characterization of these rhythms in a human subject under health and disease (in the relevant tissue) requires measurements at regular time intervals over one

available without restriction. The machine learning algorithm COFE is available via GitHub at https://github.com/bharathananth/COFE, with the published version archived on Zenodo at https://doi.org/10.5281/zenodo.15167158. The complete set of code and data necessary to reproduce all Figures and analyses presented in the manuscript is accessible through a Zenodo repository at https://doi.org/10.5281/zenodo.15152340.

**Funding:** B.A. acknowledges support from the Deutsche Forschungsgemeinschaft (DFG) grant AN 1553/2-1 (Project No. 444137814) and SCHM 3362/4-1 (Project No. 511886499). The funders played no role in study design, data collection and analysis, decision to publish, or preparation of the manuscript.

**Competing interests:** The authors have declared that no competing interests exist.

or more cycles, termed a time series [3]. Would *you* want to give repeated samples of internal tissues to quantify your circadian rhythm? It is highly unlikely that you do, and such a procedure would also contradict sound medical and ethical practice.

Therefore, we and others recently developed assays to quantify internal clock time or the *phase* of human rhythms from a single biosample [4–8]. These approaches involve discovering a small panel of circadian biomarkers using machine learning (ML) and then predicting the clock phase from levels of the circadian biomarkers in a single sample [9]. However, the discovery of biomarkers for these assays still requires time series sampling of humans, albeit a small cohort under highly-controlled conditions. This makes these assays feasible only for determining rhythm parameters in or from accessible peripheral tissue, such as blood, skin, oral mucosa or hair follicle cells [10].

How then can we quantify rhythms in human clock tissue, such as liver or kidney, that are not easily accessible? The lack of time series can be surmounted by reconstructing rhythms from post-mortem tissue samples (without time labels) from a large cohort of humans [11, 12]. These studies reconstruct a rhythm across a population (a *population* rhythm) rather than a rhythm in each subject. Nevertheless, population rhythms are still valuable as long as the circadian variation exceeds the inter-individual variation, which appears to be the case [13].

This reconstruction relies critically on algorithms to assign time labels to samples based only on the sample data, since time labels for samples might be unavailable or unreliable (e.g., time of death). To reconstruct time labels from samples, each sample must consist of multiple features (mRNA, proteins, etc.), typically measured using high-throughput methods. Patterns among these different features (or biomarkers) encode time. Scholz et al. [14] first proposed a type of nonlinear principal component analysis called circular PCA, implemented using an autoencoder with circular units, which served as the basis for the popular method CYCLOPS and its variants [11,15,16]. Talamanca et al. [12] used an expectation-maximization-based algorithm on a set of core clock genes that leverages the availability of multiple post-mortem tissues from a single donor for reconstruction. Finally, very recently, Larriba et al. [17] leveraged the known order of peak times of a small set of core features to reorder samples.

Analogously, single-cell technologies also produce data where the state of the clock in each cell is not known, i.e., each cell is not time labelled. In this context, Oscope [18] aggregates pairs of oscillating mRNA (features) in single-cell RNA-sequencing (scRNA-seq) data using a paired sine model, while Cyclum [19] builds on a circular autoencoder very similar to CYCLOPS to order cell cycle phases. Another method, Tempo [20], uses a variational Bayes approach that models the statistical properties of scRNA-seq data to achieve the same end. More recently, scPrisma [21] uses spectral computation with a cyclic topological prior to infer the rhythmic cell cycle signal.

These approaches however suffer from either or both of the following limitations. First, they typically need to be seeded with a set of features that are known or expected to cycle for the reconstruction to function well, i.e., they require some prior biological insight. Yet, this assumption on rhythmic features (based on studies on model organisms and limited human studies) may or may not hold under different conditions, e.g., when the clock is disrupted in disease. So, an approach that assigns time labels using de novo (data-driven) rhythmic features is highly desirable. Second, they rely on a two-step procedure consisting of time label assignment followed by rhythm detection that is statistically suboptimal and limits the precision in rhythmic feature detection [22].

Here, we present a data-driven approach that simultaneously reconstructs the time ordering of data and identifies de novo the list of rhythmic features that contribute to the ordering. These rhythmic features (circadian biomarkers) can be later used to assign time labels to

new samples not seen during training. Our approach, Cyclic Ordering with Feature Extraction (COFE), relies on a new variant of nonlinear PCA, called Sparse Cyclic PCA (scPCA), combined with a novel unsupervised cross-validation technique to fix the optimal sparsity level. We used COFE to reveal and compare for the first time circadian rhythms in vivo in 11 different human cancers based on the multi-omic data from The Cancer Genome Atlas (TCGA) database. Our study finds widespread rhythms at the transcript and protein levels in tumor tissue that are uncoupled with healthy clocks, but coupled with the cell cycle. COFE thus opens the door to exploring human circadian rhythms in internal tissues within their natural systemic context.

## Results

### COFE reconstructs time labels and predicts rhythmic features with high accuracy

To develop a suitable approach to assign time labels to samples, we relied on our observation that high-dimensional rhythmic time series data (with time labels) projected on a suitable plane produces a closed curve, where the samples are ordered in time (within one cycle) [5, 7,13,23]. We formulate this problem as a search for a low-dimensional elliptical approximation of the data that can be solved efficiently using a simple and fast iterative algorithm (Materials and Methods – Theory, S1 Text). Sparse Cyclic PCA (scPCA) finds this circular projection in data without time labels, while also selecting rhythmic features to aid in the circular projection. This procedure can thus be equally cast as dimensionality-reduction or manifold-learning. scPCA combined with a novel cross-validation (CV) scheme for unsupervised learning (Materials & Methods – Implementation) automatically learns the optimal sparsity parameter from the data (S1A-B Fig), reconstructs the circular projection (S1C Fig), assigns time labels to samples (S1D Fig) and identifies rhythmic features in the data (S1E Fig). We implemented scPCA in a Python package named COFE (Fig 1, S2 Text, https://github.com/bharathananth/COFE, https://doi.org/10.5281/zenodo.15167158), and benchmarked it on synthetic and biological (transcriptomic) data.

On synthetic data, COFE performed consistently well and its performance improved with increasing signal-to-noise ratios (SNRs), larger number of samples and higher fraction of rhythmic features in the data (S1F-H Fig). We quantified both COFE's ability to predict time labels of samples and its ability to accurately (precision) and exhaustively (recall) identify the true rhythmic features in synthetic data containing both rhythmic and arrhythmic features with additive noise (S3 Text). COFE predicted the time labels of samples in the data with high accuracy that improved with lower noise, higher number of samples and rhythmic features (S1F Fig). COFE achieved perfect precision except at the smallest sample size (S1G Fig) and approached perfect recall with "better" and "larger" data (like the ordering performance) (S1H Fig).

The CV scheme serves to find the optimal sparsity parameter value as well as quantify the consistency and quality of the temporal ordering. Successful ordering is signalled by COFE with a clear minimum in the CV output (S1B Fig). The minimum imputation error achieved during CV is smaller at higher SNRs, larger sample sizes and larger fractions of rhythmic features (S2A Fig), where better ordering performance was achieved. The optimal sparsity grows with larger and more rhythmic datasets, roughly as the square-root of the number of rhythmic features in the data (S2B-C Fig). When there are no rhythmic features in the data, the output of CV increases monotonically and contains no non-trivial minimum (S2D Fig). Finally, sample times restricted to a fraction of the cycle (as might occur with samples only collected

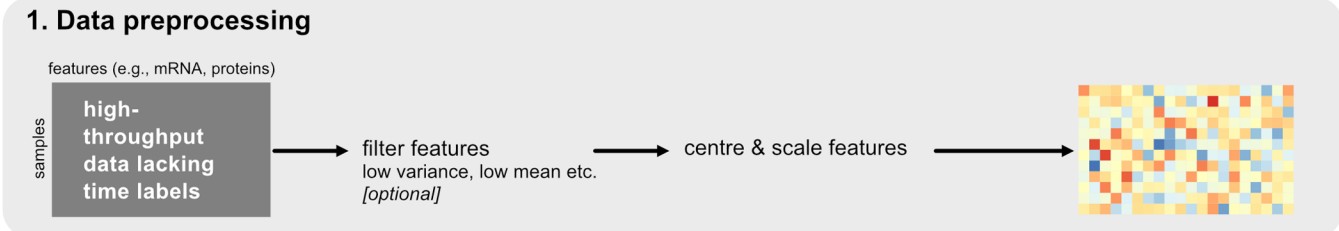

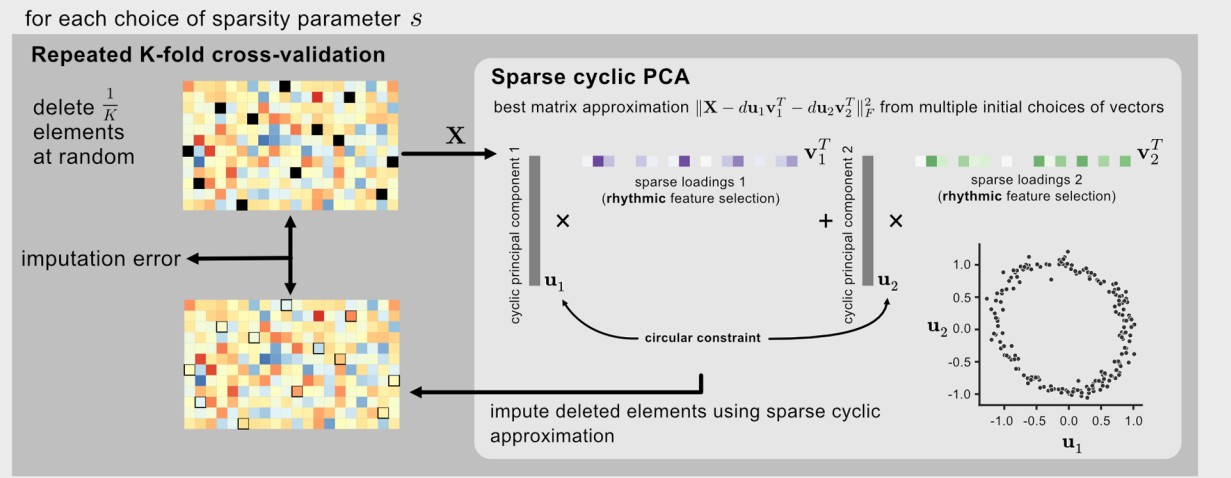

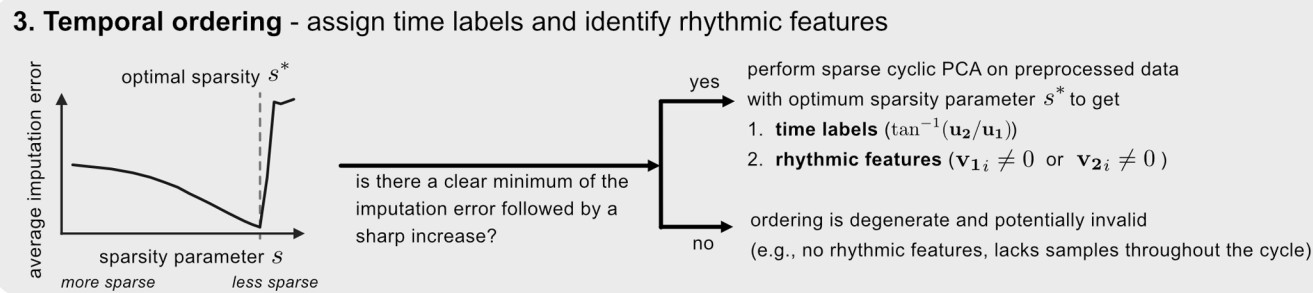

**Fig 1. Workflow of Cyclic Ordering with Feature Extraction (COFE).** Data without time labels are first optionally pre-filtered and then standardized. A novel cross-validation approach finds a value for the unknown sparsity parameter that guarantees good temporal ordering. Finally, COFE both assigns time labels to the samples and produces a list of rhythmic features in the data.

during a part of the circadian cycle), are similarly flagged by a CV output with no non-trivial minimum (S2E-H Fig).

COFE provides high resolution temporal ordering and high precision in finding rhythmic features across a wide range of data settings compared to state-of-the-art methods (S3 Fig). We compared COFE against autoencoder-based (CYCLOPS [15]), graph-based (PAGA [24]) and spectral analysis-based (scPrisma [21]) methods on synthetic data (S3 Text). However, unlike COFE, none of the methods can both assign time labels and identify rhythmic features; the first two do not extract rhythmic features, while the last one does not assign time labels. While COFE and CYCLOPS provided highly accurate time labels across a range of

SNRs, scPrisma and PAGA had inferior performance at low and high SNRs, respectively (S3A Fig). COFE excelled in identifying rhythmic features without errors (perfect precision), which was unmatched by the other methods (S3B Fig). The precision of CYCLOPS and PAGA was limited by Type I errors in the cosinor rhythm detection used for rhythmic feature identification [25], whereas scPrisma had very poor performance at low SNRs. PAGA and CYCLOPS exhaustively recovered rhythmic features (along with false positives), which COFE and scPrisma achieved at sufficiently large SNR, number of samples or fraction of rhythmic features.

On biological time series (labelled) data, COFE's performance was only a little worse than the corresponding supervised approach (S4 Fig). In these data (S1 Table), we only evaluated COFE's ability to predict time labels, since the ground truth regarding the true rhythmic features is unavailable. COFE was able to find the two-dimensional projection for mouse liver time series data (S4A-B Fig) and predicted time labels to within the sampling resolution of the data (S4C Fig). Time labels of longitudinal human time series data from blood monocytes [23](S4D Fig) and skin [13](S4E Fig) could also be predicted almost as accurately as the corresponding supervised approach (using Zeitzeiger) [5] with accuracies of 0.75 h and 1.2 h, respectively. Finally, COFE was able to predict both the time labels and the underlying period of the clock on non-mammalian time series data, such as from different strains of the malaria parasite [26] (S4F Fig).

Therefore, we are confident in COFE's ability to reconstruct underlying rhythms and identify rhythmic features in high-dimensional biological datasets, when present, and to flag degenerate situations (no rhythms, poor sampling over a cycle) by means of the CV output.

## Widespread *in vivo* population rhythms in human adenocarcinomas

To investigate human rhythms under heretofore inaccessible conditions, we leveraged the high-quality transcriptomic data from human cancer biopsies assembled in The Cancer Genome Atlas (TCGA) database [27]. These data are a collection of patient-derived single samples from a variety of human cancers. We restricted our attention to adenocarcinomas (ACs), which are cancers of epithelial origin that develop in organs and glands and account for 80 to 90 percent of all cancers [28]. We further limited our study to ACs with sufficient large number of samples (>250) within a single histological type (S2 Table), which ensured homogeneity of the samples and comparability of the different ACs (S1 Text). This resulted in a collection of 11 different ACs – ductal breast (BRCA), clear cell (KIRC) and papillary (KIRP) kidney, prostate (PRAD), ovarian (OV), uterine (UCEC), lung (LUAD), colon (COAD), thyroid (THCA), bladder (BLCA) and liver (LIHC) cancers. Samples from each AC were evenly balanced between sexes, except for the four sex-specific ACs (BRCA, PRAD, OV, UCEC) (Fig 2A).

COFE predicted time labels and identified rhythmic features (here genes) for the samples from each AC independently and reproducibly (S1 Text). Surprisingly, COFE reconstructed rhythmic gene expression profiles of 1000s of genes in each AC with the peak time of expression distributed throughout the day (S5 Fig, S6A Fig, S3 Table). The CV outputs with clear minima confirmed that rhythms indeed were present in each AC (S6D Fig). The number of rhythmic genes identified by COFE varied greatly between the different ACs, ranging from about 2000 in uterine AC to over 6000 in ovarian AC out of about 13000 expressed genes in each AC (Fig 2B). When restricted only to the rhythmic genes with at least a two-fold peak-to-trough amplitude, the variability in the number of rhythmic genes across ACs was greatly reduced (S6B Fig). The number of rhythmic genes were independent of the number of samples in the AC ($r_s(9) = 0.04$, $p = 0.92$). In general, ACs with more rhythmic genes not only had

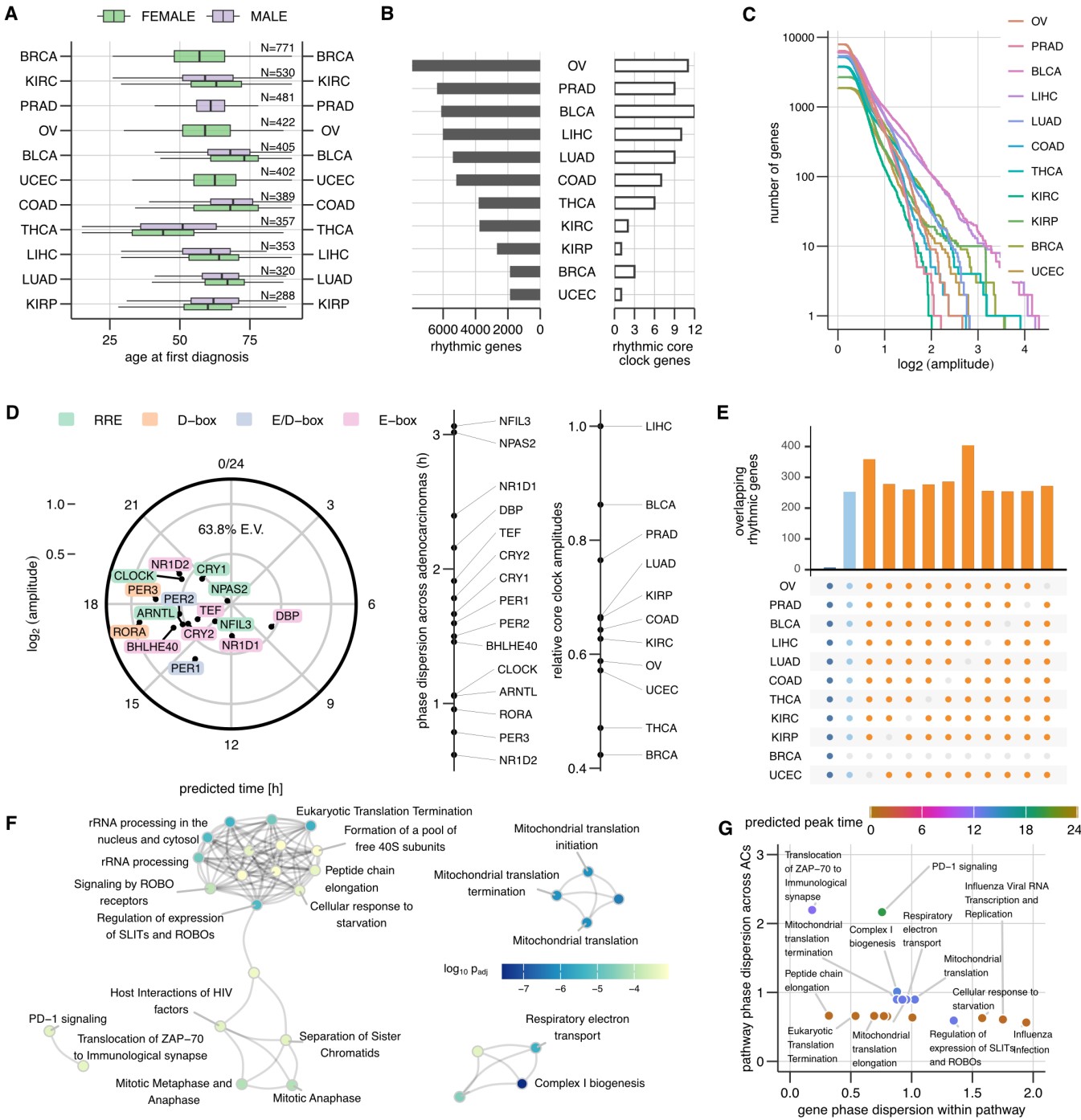

**Fig 2. Circadian population rhythms in human adenocarcinomas (ACs).** (A) Demographics of patients, whose biosamples provided data for training COFE. (B) Number of rhythmic genes and rhythmic core clock genes identified in each AC. (C) Cumulative distribution of the rhythmic gene amplitudes in different ACs. (D) Consensus amplitude and peak times of 15 commonly expressed core clock genes across the ACs (left). The dispersion of peak times of clock genes from the consensus arrangement across the ACs (middle). The relative strength of core clock network in different ACs (right). (E) Intersection of rhythmic gene sets across 11, 10 and 9 different ACs. (F) Reactome-based gene-set enrichment analysis of the common rhythmic genes. (G) Variability of peak times of genes within a pathway compared against the variability of peak times with respect to the core clock in different ACs. The data underlying Figure panels can be found in S1 Data.

larger median amplitudes, but also achieved higher amplitudes of gene expression (Fig 2C). The exception to that was BRCA, where the number of high-amplitude rhythmic genes (>4-fold peak-to-trough) was comparable to the ACs with numerous rhythmic genes.

Thus, COFE revealed widespread and heterogeneous in vivo rhythms in human cancers.

## Core clock is rhythmic in human adenocarcinomas

Due to the absence of an inherent timescale in the biopsy-derived transcriptomic data, COFE alone cannot ascribe a period to the reconstructed rhythms. To infer this period, we investigated a set of 20 core clock genes that constitute the fundamental rhythm generating mechanism of the cell autonomous clock. Many of the 15 expressed core clock genes were also identified as rhythmic in ACs (Fig 2B, S6C Fig, S3 Table), although the expressed core clock genes were not statistically more rhythmic than other output genes. This is indicative of a rhythmic core clock network in several ACs.

COFE's time label assignments are consistent within each AC, but time labels from different ACs do not share a common time reference. We therefore constructed the consensus relationships between the peak times of the 15 commonly expressed core clock genes across 11 ACs in the form of an 'eigen-core clock' using complex SVD [12]. We used this consensus core clock arrangement to define a time reference for each AC. The arrangement of peak times was reasonably well conserved across ACs (and explained 64% of the inter-AC variation) (Fig 2D). This de novo analysis arranged peak times of most core clock genes in the expected order based on the response elements in their promoters (E-box → E-box/D-box → D-box → RRE) and the peak times spanned half a cycle like in a healthy functional clock [7,12]. The core clock genes with atypical (disrupted) peak times (circadian phase) were E-box targets NR1D2, TEF, BHLHE40 and E-box/D-box target PER2, which all peaked later in the day than expected (phase delayed).

Simultaneously, the analysis also revealed the strength (amplitude) of distinct clock genes across the different ACs. The D-box targets RORA and PER3 had the largest amplitudes (more than threefold peak-to-trough), while RRE targets NPAS2, NFIL3 and CRY1 had the smallest amplitudes (approximately twofold peak-to-trough) (Fig 2D). The amplitudes in the consensus core clock combine the amplitudes in each AC and degree of peak time consistency of the genes across ACs. The median amplitude of each clock gene across ACs correlated strongly with consensus amplitudes ($r_p(13) = 0.86, p < 0.001$). NR1D2 not only peaked aberrantly about 8h late, but had extremely consistent peak times in the ACs. Core clock genes with higher amplitudes deviated less from the peak time in the consensus arrangement (Fig 2D, $r_s(13) = -0.86, p < 0.001$). Thus, clock genes with larger consensus amplitudes both had higher amplitudes in each individual AC and generally adhered to the consensus peak time across ACs.

We finally evaluated the relative strength of this consensus core clock arrangement in the different ACs. The core clock strength ranged from the strongest in the liver AC and weakest in the breast cancer AC, which was only half as strong. Interestingly, the ordering based on core clock network strength correlated only weakly with an ordering of ACs based on the number of rhythmic genes ($r_s(13) = 0.58, p = 0.066$).

In summary, there is a functional yet disrupted core clock that is conserved across different cancers, but the similarity of the conserved peak times of the core clock in the cancers to healthy tissue lets us conclude that these rhythms are indeed circadian.

## Key cellular processes are enriched for rhythmic genes

To identify the processes enriched for rhythmic genes in different ACs, we performed gene set enrichment analysis [29] using the Reactome database (reactome.db-v1.88.0)[30] with genes scored based on the number of ACs in which the gene was rhythmic. Although only 7 rhythmic genes were shared between all the ACs, the ACs excluding BRCA shared 250 rhythmic genes (Fig 2E). Moreover, the overlap across different 9 out of 11 ACs again excluding BRCA was also 200–300 genes, which were all highly significant (Fisher Exact Test, $p < 0.001$). The genes with higher scores (i.e., rhythmic in more ACs) were highly enriched in Reactome pathways that fell into four Reactome pathway clusters – mitochondrial translation, respiratory electron transport, mitotic cell cycle, and adaptive immune system (Fig 2F). Respiratory electron transport and mitochondrial translation were the most highly enriched categories ($p < 10^{-6}$) making cellular metabolism the most rhythmic process in ACs.

We next quantified the consistency in peak times of rhythmic genes within the enriched Reactome pathways across different ACs. We computed two metrics that measured the coherence of gene expression peak times within a pathway and the consistency of the peak arrangement across ACs (Fig 2G). This resulted in three types of pathways – pathways with highly synchronized rhythmic genes peaking at the same time with respect to the core clock network (e.g., translation), pathways with synchronized rhythmic genes but the gene expression peaks in each AC at different times with respect to the core clock (e.g., programmed cell death) and pathways with neither synchronized rhythmic gene expression nor fixed times with respect to the core clock in different ACs. The peak activity of genes in respiratory electron transport (with respect to the core clock) varied by utmost 6h across ACs and the genes within this category were synchronized except for a few subunits of respiratory complex I and complex IV.

The identified rhythmic processes are therefore fundamental and highly conserved across the cancers.

## Clocks in tumors are uncoupled from clocks in healthy tissue

We thus far presented the results using the core clock in each AC as a reference to define the peak time of rhythmic gene expression. Since the core clock in different tissues in the circadian network typically are not phase-aligned, we would like to determine gene expression with respect to the network (at the organismal level). We therefore first considered the distribution of predicted time labels in each AC over one circadian cycle. If clocks in the ACs are phase-locked to the circadian system, which in turn is entrained to the light-dark cycle, and assuming these tissue biopsies were gained from procedures conducted during the day (standard in most countries), we expect the predicted time labels to be restricted to a fraction of the circadian cycle. Surprisingly, predicted time labels lay throughout the circadian cycle in all ACs (Fig 3A). This is also consistent with the clear minima observed in the CV output for all ACs (S6D Fig). Although the biosamples in TCGA were obtained from several countries, the predicted time labels were only weakly associated with the origin of the samples (MANOVA effect size $\eta^2 < 0.06$, S7A Fig), except for a strong effect (MANOVA effect size $\eta^2 = 0.26$) in COAD, and medium effect in UCEC and LIHC (MANOVA effect size $0.06 < \eta^2 < 0.14$). Simultaneously, we also verified that the sex of the subjects was not associated with time labels in the seven non-sex-specific ACs (MANOVA effect size $\eta^2 < 0.01$) and the staging of the cancers in these patients also only had an effect on the time labels in BRCA, UCEC and KIRP, where the effect was not strong ($0.07 \leq \eta^2 \leq 0.1$).

Next, we compared the predicted time labels for tumor tissue samples with the patient-matched healthy tissue samples that were available for 10 of the 11 ACs for a small fraction

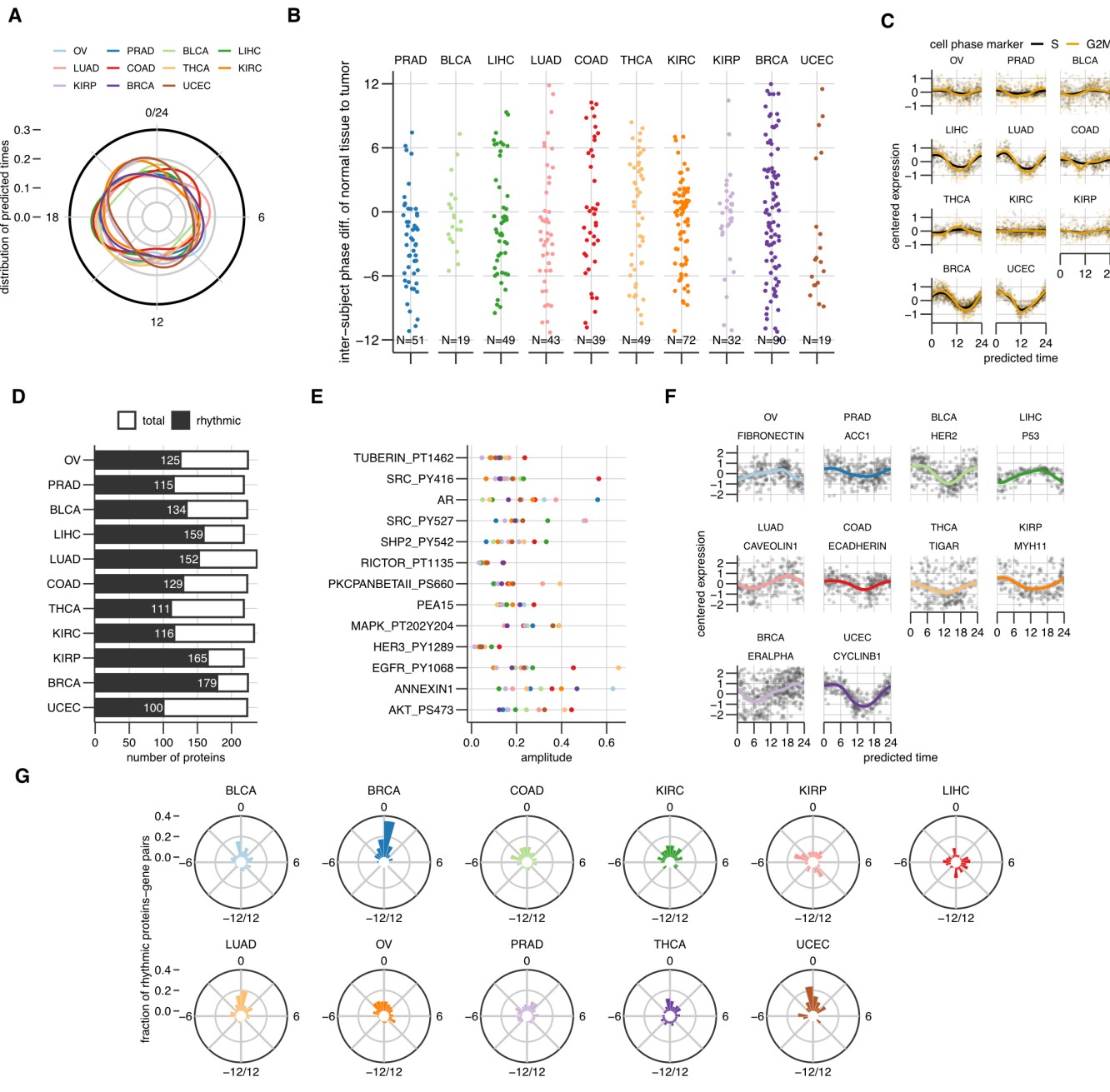

**Fig 3. Coupling of the circadian clock to the cell cycle and the proteome.** (A) Distribution of predicted time labels for the samples in each AC. (B) Difference in time labels predicted by COFE between cancer and patient-matched healthy samples, where both are available. (C) Scores for cell cycle phases (S and G2/M) in the different ACs. The raw and LOESS-smoothed scores per sample are shown. (D) The number of rhythmic and total measured proteins in each AC. (E) The amplitude of rhythmic protein expression (color coded as in (A)) for proteins including modified forms that are rhythmic in at least 10 of the 11 ACs. (F) The raw data and LOESS-smoothed means for the rhythmic protein with the highest amplitude in each AC. (G) Distribution of the difference in peak times of expression of rhythmic proteins and their corresponding rhythmic gene. The data underlying Figure panels can be found in S1 Data.

of the original patients (Fig 3B). The difference between the predicted time labels for healthy and tumor tissue was distributed throughout the circadian cycle in most ACs suggesting that the time differences are almost random. As a negative control, we compared the difference between predicted time labels for patient-matched tumor tissue not used to train COFE (S7B Fig). As expected, these differences clustered around zero.

Based on both these observations, we conclude that the clocks in tumors do not have a fixed timing relationship with clocks in healthy tissue.

## Circadian clock is coupled with the cell cycle in several ACs

Our previous analysis suggested that tumor clocks are not synchronized with the healthy clocks within the same tissue. We wondered whether the tumor circadian clock might be instead coupled to other rhythmic processes in the cell, such as the cell cycle that was enriched in our Reactome analysis (Fig 2F). To test this, we computed gene expression-based scores for cell cycle phases (S and G2M) (Fig 3C)[31]. Both the scores oscillated over the circadian cycle in all ACs (cosinor, $p < 0.003$) except KIRC ($p = 0.39$). The largest amplitude rhythms in the scores were in BRCA, UCEC and LUAD. Surprisingly, the two least rhythmic ACs (BRCA and UCEC) both showed robustly rhythmic scores. These marker scores can also predict the predominant cell cycle phase of cells in these (bulk) tissue samples. The ACs with rhythmic scores indeed show distinct time-of-day dependent cell cycle phases of the constituent cells, with G1 and G2/M phases separated in time (S7C Fig). So, the two important cyclic processes in the cell are synchronized in several human cancers.

## Protein expression is rhythmic, even in ACs with few rhythmic genes

Since COFE assigns time labels to samples, we can also assign time labels to measurements from other assays run on the same samples. The Cancer Proteome Atlas (TCPA) database contains the protein expression of a subset of the proteome (258 proteins and modified forms) important in cancer measured using Reverse Phase Protein Array on the same tumor samples in TCGA. We used a cosinor approach [25] to find the rhythmic proteins, including modified forms, in the different ACs (S4 Table). 237 proteins were rhythmic in at least one AC. Moreover, 13 proteins were rhythmic in only one AC, while three were rhythmic in all. Unlike the transcriptome, at least half the measured proteins were rhythmic in all ACs (Fig 3D). Unexpectedly, BRCA had the most rhythmic proteins despite BRCA having among the fewest rhythmic genes and also the least overlap of rhythmic genes with other ACs. 13 proteins were rhythmic in 10 out of 11 ACs (Fig 3E). The nuclear receptor AR, phosphorylated tyrosine kinase SRC and phosphorylated Tuberin were rhythmic with robust amplitudes in all ACs.

We associated the rhythmic proteins from TCPA to the associated pathways defined in [32] (S4 Table). Members of multiple key cancer pathways including activated forms were rhythmic in at least 9 of the 11 ACs: apoptosis (PEA1S, TRANSGLUTAMINASE), cell cycle (ARID1A, BAP1C4, CHK2), epithelial–mesenchymal transition (CLAUDIN7, ECAD-HERIN), MAPK signaling (MAPK_PT202Y204, MEK1, P38_PT180Y182), metabolism (ACC1, FASN, G6PD), PI3K/AKT signaling (AKT_PS473, PI3KP85) and receptor tyrosine kinase signaling (EGFR_PY1068, HER2_PY1248, HER3_PY1289, SRC_PY416). Nonetheless, the protein with the highest amplitude in each AC varied and sometimes also included proteins that were rhythmic in only a few of the ACs, e.g., P53 and HER2 (Fig 3F).

We next related the rhythmicity of the proteins to the rhythmicity of their underlying genes. The genes of rhythmic proteins were significantly more likely to be rhythmic than

arrhythmic proteins ($p < 0.001$ for each AC, Fisher Exact test) except in the two most rhythmic ACs, OV and PRAD. Similarly, proteins of rhythmic genes were more likely to be rhythmic than proteins of arrhythmic genes ($p < 0.02$ for each AC, Fisher Exact test). We also compared the peak expression times of rhythmic proteins and their corresponding rhythmic genes (Fig 3G). The peak of rhythmic protein expression was generally synchronized with the peak of rhythmic gene expression for most ACs (average difference ±1.5h). The notable exception here was LIHC, where the rhythmic proteins peaked on average 6h after their respective rhythmic gene.

In summary, there was broad agreement between the independent analyses that identified transcript and protein rhythms.

## A large, diverse array of small molecule drugs target rhythmic genes in tumor tissue

If the rhythmic genes include targets of approved and experimental cancer drugs, time-of-day changes in the target could be potentially leveraged to improve drug efficacy (chronotherapy)[33]. We identified rhythmic gene targets of a list of 481 FDA-approved, clinical candidates and putative small molecules that target key cancer pathways or processes from the Cancer Therapeutics Response Portal [34,35] (S5 Table). 51 of the 53 FDA-approved drugs in the database targeted a rhythmic gene on average in around five different ACs and included several that had rhythmic targets in most ACs (Fig 4). The targets of two widely-used chemotherapeutic drugs, Teniposide and Gemcitabine, were rhythmic in all but one AC. Half the rhythmic FDA approved drug targets had better than 3-fold peak-to-trough amplitude. In 30% of cases, FDA approved drugs had more than one rhythmic target in an AC. Moreover, the distribution of FDA approved drug target amplitudes were similar across the different ACs with a few exceptions. Interestingly, targets of tamoxifen and fulvestrant in BRCA had the largest amplitudes, despite BRCA being one of the least rhythmic ACs. The peak time of the FDA approved drug target expression was in general similar across ACs in which they were rhythmic except for tacrolimus, microsporic, thalidomide and dasatinib that differed by up to 12h between ACs (Fig 4B).

91 of the 95 clinical candidates targeted rhythmic genes in some AC. Although clinical candidates also typically had rhythmic targets in about five ACs like the FDA approved drugs, five candidate drugs (PHA-793887, SNS-032, XL765, dinaciclib, rigosertib) had rhythmic targets in all ACs. Amplitudes of clinical candidate drug targets were generally smaller than the FDA approved drugs (Fig 4A). Similar to the FDA approved drugs, candidate drug targets in BRCA and UCEC (the least rhythmic ACs) had the highest median amplitude of 2-fold peak-to-trough. Clinical candidate drugs too could have targets with greatly different peak times (e.g., AZD4547, tosedostat), but the candidate drugs with rhythmic targets in all tissues had consistent phase across ACs.

Almost all putative cancer drugs in the database (248/267) targeted a rhythmic gene (S8 Fig). These putative drugs were largely similar to the clinical candidates in the average number of ACs with rhythmic targets and number of drugs with rhythmic targets in all ACs. Combinations of drugs involving selumetinib targeted rhythmic genes most broadly (S8A Fig). Putative drug targets in BRCA again and LIHC had the highest-median amplitude of 3-fold peak-to-trough. Compared to the other two drug classes, peak times of putative drugs were consistent across ACs, although they included some drugs with diversely peaking targets (selumetinib+navitoclax) (S8B Fig).

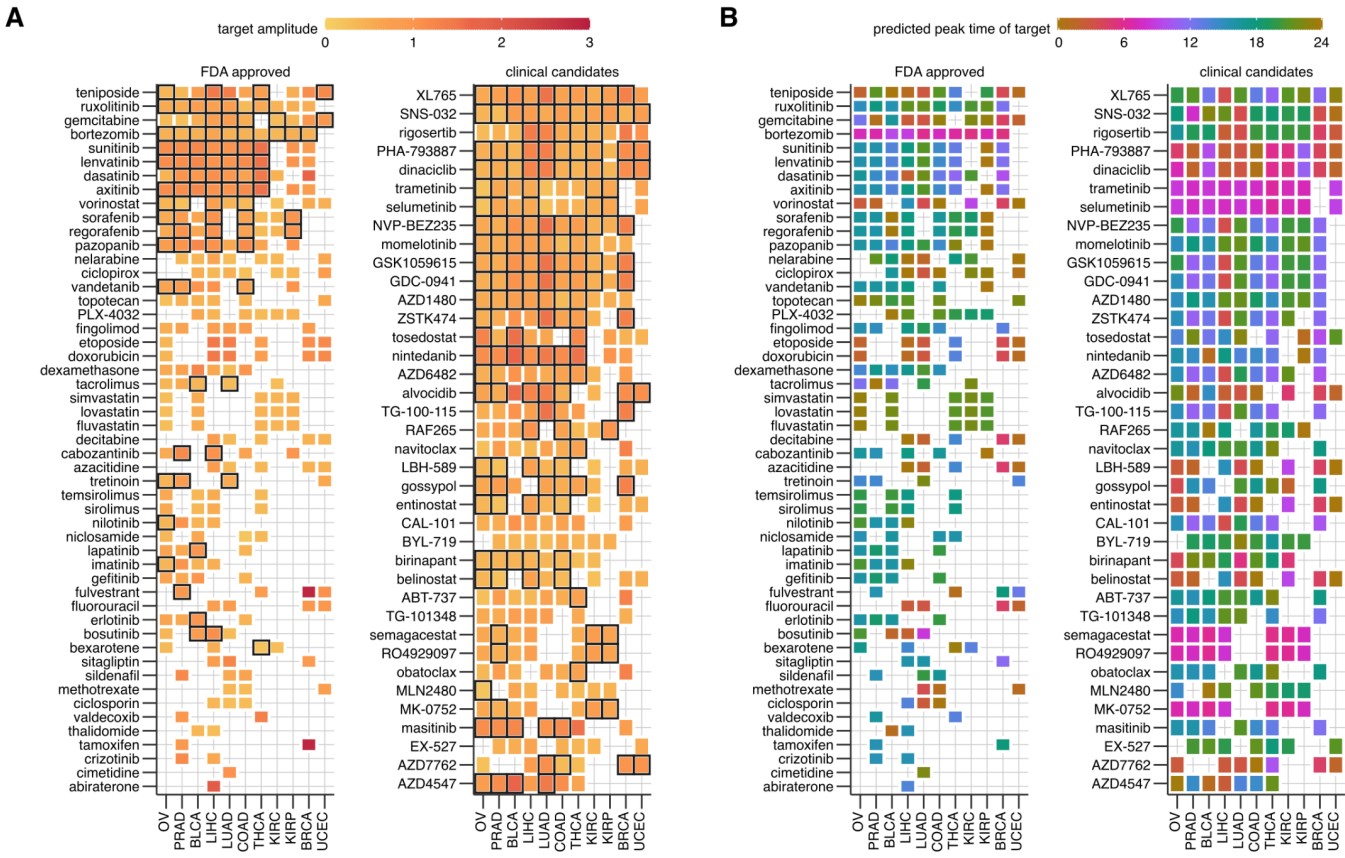

**Fig 4. Rhythmic targets of FDA-approved and clinical candidate cancer drugs.** The $\log_2$-fold amplitude (A) and peak time relative to the core clock (B) of the largest rhythmic target gene in each AC of FDA approved and clinical candidate drugs that target cancer pathways or processes. All FDA-approved drugs and clinical candidates with rhythmic targets in at least 7 ACs are included. Drugs with multiple rhythmic gene targets in an AC are boxed in black. The data underlying Figure panels can be found in S1 Data.

## Discussion

In this work, we addressed the principal challenge to studying rhythms in humans, namely that repeated time series sampling is necessary to study rhythms and these are difficult to obtain from most clock-relevant tissues in humans. We instead chose to reconstruct *population* rhythms from high-throughput (omics) snapshot data (one sample per individual) across a homogeneous cohort rather than quantifying rhythms in an individual, similar to previous studies [7,12,15]. Such snapshot data are much easier to obtain from databases or directly from biopsies and post-mortem biosamples, but lack time labels specifying the sampling time or internal circadian time of the individual. Therefore, we developed an ML algorithm named COFE that can not only assign missing time labels to snapshot samples, but, unlike previous approaches [12,15], can also identify rhythmic biomarkers (features) de novo in the data. COFE identifies only truly rhythmic features as biomarkers and predicts time labels with an accuracy comparable to supervised approaches, such as Zeitzeiger [5]. The CV in COFE ensures that the sparsity parameter is data-driven and, more importantly, is able to flag spurious orderings that might arise due to a lack of rhythms or samples spanning only a part of a cycle, which arises in many human circadian contexts.

In humans, steadily accumulating epidemiological evidence has linked increased cancer risk to environmental disruption of the circadian clock in multiple cancers, including breast and prostate cancers [36,37]. Investigations into the molecular relationships in humans between the circadian clock and cancer using cell lines have however yielded heterogeneous results [38]. Even whether cancerous cells in peripheral clock tissues possess cell autonomous rhythms is unclear and very dependent on cell lines (subtype, genetic background) and conditions [39,40]. Moreover, studies on cell lines overlook the systemic circadian inputs that peripheral tissue receive in vivo. Furthermore, translating insights from animal models to humans can be challenging. We therefore leveraged snapshot transcriptomic data from large cohorts in the TCGA database to study in vivo rhythms in multiple human cancer tissues for the first time.

We discovered robust rhythms in 1000s of genes in all ACs, but to varying extents, which converged when restricted to only large amplitude rhythms. Interestingly, the hormone-dependent cancers- the most common cancers- accounted for the extremes in the number of rhythmic genes, with ovarian and prostate cancer containing more than 40% rhythmic genes, while breast and uterine cancer had less than 15%. A recent study [16] that used CYCLOPS 2.0 to reorder the luminal A subset of BRCA data (193 samples) also found about 1100 rhythmic genes, and the high amplitude genes from that study overlapped highly significantly (226/567, $p < 0.001$) with our own. Barring technological advances that allow non-invasive time series sampling of internal tissue, ethical constraints preclude direct validation of these rhythms in vivo. Nevertheless, the rhythms we found are bona fide patterns and not artifacts because of the following reasons. First, the CV in COFE showed a clear minimum in the imputation error for all ACs, indicating the presence of rhythms and samples spread throughout the circadian cycle. Second, there are large statistically significant overlaps in the rhythmic genes discovered in each AC, which were all obtained from independent COFE analyses. Third, the predicted time labels of the samples do not represent ordering of gene expression based on another factor, such as origin of the sample, cancer stage or sex. Fourth, we found consistency in expression phase between rhythmic genes and the rhythmic proteins they encode.

The TCGA data do not include direct temporal information to estimate the period of the discovered rhythms. Nevertheless, multiple lines of evidence support the conclusion that these are indeed *circadian* rhythms. First, COFE's de novo reconstruction classified as rhythmic many, but not all, known core clock genes in all the ACs; luminal A BRCA samples too only exhibited rhythms in seven core clock genes according to a recent study [16]. Second, the peak expression of the core clock genes were arranged in the known sequence based on the response elements in their promoters in all ACs. Third, this arrangement of peak expression was mostly conserved across different independently-reordered ACs. The key difference between the core clock in these cancers and healthy tissue was a significant delay in the peak expression times of core clock genes NR1D2, PER2, TEF and DEC1. The nuclear receptor NR1D2 was delayed almost 8h compared to is homolog NR1D1, which is consistent with similar observations in live cell imaging of hepatocellular carcinomas in mouse liver (K. Padmanabhan, personal communication). This discordance between NR1D1 and NR1D2 is a possible source of the small amplitudes in RRE-targets NPAS2, NFIL3, and CRY1 in the cancers. The D-box targets, PER3 and RORA, had the largest amplitudes as observed in healthy human tissue [12] and simultaneously the smallest variability in peak times across ACs. This trend that amplitudes and peak time variability were inversely correlated held across all clock genes. Since peak time estimates are poorer with smaller amplitudes, this observation supports the view that the consensus peak time arrangement might be even more conserved. Previous work on clock disruption in cancer using TCGA data do not shed light on these aspects, since

they were either based on the significant changes in the correlation structure [41] or changes in average expression between tumor and healthy matched tissue [42] of a small set of core clock genes.

The commonly rhythmic genes in the different ACs were all involved in Reactome pathways associated with the Hallmarks of Cancer [43]. While it is known that processes, such as cellular metabolism, signaling and mitosis, are misregulated in cancer, it is unclear how robust rhythms in these processes might aid cancer progression. The peak times of some of these processes are also heterogeneously timed with respect to the core clock in each AC. Since we do not have access to the systemic circadian clock in these patients, we cannot assess whether these rhythms have evolved to be in-phase or out-of-phase with the circadian clock network. However, the lack of a stable relationship between predicted times of healthy matched tissue and tumor tissue suggests that, although the ACs appear to possess robust rhythms, these circadian rhythms are uncoupled from the systemic clock in the subject. This hypothesis is also supported by the suggestion from CV that the samples were indeed collected around the clock, which considering standard clinical schedules in most countries, is unlikely. A similar lack of correlations between predicted times of healthy and tumor samples was observed recently in a subset of BRCA samples [16]. We also inferred strong coupling between the cell cycle phases and circadian clock in many ACs, which predisposed cell cycle transitions to certain clock times. Previous in vitro studies have proposed uni- and bidirectional link between the two [44]. We can further speculate that this coupling between the cell cycle and the clock might override the coupling between the tissue clock and systemic clock, leading to apparent free running clocks in tumors.

Assigning time labels to samples based on transcriptomic data allows us to do much more than study gene expression rhythms. We also used the time labels for multi-omic data integration to study temporal proteomic patterns. About 50% of measured cancer-relevant protein and protein forms were de novo rhythmic in each AC, with some proteins rhythmic in all ACs. In line with the Reactome enrichment of the rhythmic genes, proteins and their activated forms in seven key cancer pathways (apoptosis, cell cycle, epithelial-mesenchymal transition, MAPK, PI3K/AKT, metabolism and receptor tyrosine kinase) were rhythmic in almost all ACs, which is also consistent with insights from gene expression signatures [45]. These observed rhythms raise questions about the interpretation of the observed misregulation of these pathways in cancer. In addition, taking these rhythms into account could help stratify individuals better, especially if levels of these proteins are used as prognostic markers for cancer progression. The rhythmicity of genes and their protein product were also strongly associated in their rhythmicity and peak phase of expression. The consistent insights from the transcript and protein levels further validate the ordering produced by COFE.

From a therapeutic perspective, the widespread rhythms we observed in the transcriptome and proteome of tumor tissue presents the possibility to leverage time to improve and optimize the effect of anti-cancer drugs. Remarkably, most analyzed small molecule drugs targeted a rhythmic gene in at least one AC, indicating that time of administration ought to be considered (depending on the half-life of the drug) and in relation to both clocks in healthy [11] and tumor tissue, which appear to be decoupled. Half the currently used FDA-approved cancer drugs targeted genes with better than 3-fold peak-to-trough amplitude. The peak time of expression of the drug targets however could be quite drug- and AC-dependent, which must be considered for drugs used against multiple ACs and when treatment with drug combinations. The hormone-dependent cancers BRCA and UCEC possessed the highest amplitude in drug targets and might be prime candidates to investigate chronotherapies.

Our novel ML algorithm COFE that enabled the study of de novo rhythms in cancer has significant advantages over other unsupervised methods [12,15,16]. These approaches first

recover the time labels and then identify rhythmic features by means of (cosinor) regression using the predicted time labels. This two-step procedure using ML-predicted covariates is known to be statistically suboptimal [22]. In contrast, COFE directly provides interpretable outputs by simultaneously estimating time labels and identifying biomarkers that contribute to the ordering. It does this via a novel variant of PCA called sparse cyclic PCA (scPCA), which can be viewed as a generalization of cosinor regression, where the sinusoidal features are estimated together with the regression coefficients (S1 Text). We combined this with a powerful cross-validation method for choosing the optimal sparsity level (Fig 1), enabling robust automated identification of rhythmic biomarkers. The scPCA formulation also allows COFE to be enhanced using a range of PCA-related techniques to address open issues, such as confounders, rhythmic features at multiple frequencies and missing values in the data.

It has become firmly established that daily temporal changes ought to be considered in all aspects of biology and medicine due to the ubiquity of circadian rhythms [1]. But, this endeavor is hampered by the need for time labels on recorded data. Here we used ML to recover time labels in a data-driven manner, which allows us to forgo the need for time series measurements and to augment data without time labels with this important covariate, i.e., "time". Although we showcased COFE on human data, our approach is equally effective in animal-model studies, where it can be used to better adhere to 3R principles and reduce the number of new animal studies that need to be performed. On the whole, our approach will greatly expand our ability to study circadian rhythms in humans and facilitate translation of experimental paradigms to the clinic.

## Materials and methods

### Theory

The $p$ features measured at $N$ (unknown) points in time are assembled in a data matrix $\mathbf{X} \in \mathbb{R}^{N \times p}$. We approximate the matrix $\mathbf{X}$ along two principal components (PCs) $\mathbf{u}_1 \in \mathbb{R}^{N \times 1}$ and $\mathbf{u}_2 \in \mathbb{R}^{N \times 1}$ with corresponding loading vectors $\mathbf{v}_1 \in \mathbb{R}^{p \times 1}$ and $\mathbf{v}_2 \in \mathbb{R}^{p \times 1}$. To achieve a variance interpretation for the projection, the data matrix $\mathbf{X}$ is column-centered, i.e., $\sum_i \mathbf{X}_{ij} = 0$ for each $j$. As such, this is a standard rank-2 reduction of the matrix $\mathbf{X}$, like PCA:

$$\min_{\mathbf{u}_1, \mathbf{v}_1, \mathbf{u}_2, \mathbf{v}_2, d} \left\| \mathbf{X} - d\mathbf{u}_1\mathbf{v}_1^T - d\mathbf{u}_2\mathbf{v}_2^T \right\|_F^2 \tag{1}$$

$$= \max_{\mathbf{u}_1, \mathbf{v}_1, \mathbf{u}_2, \mathbf{v}_2} 2d\mathbf{u}_1^T\mathbf{X}\mathbf{v}_1 + 2d\mathbf{u}_2^T\mathbf{X}\mathbf{v}_2$$

$$- d^2\|\mathbf{u}_1\|_2^2\|\mathbf{v}_1\|_2^2 - d^2\|\mathbf{u}_2\|_2^2\|\mathbf{v}_2\|_2^2 - 2d^2\mathbf{u}_1^T\mathbf{u}_2\mathbf{v}_1^T\mathbf{v}_2.$$

We refer to the objective function in the first line of Eq (1) as the *matrix approximation error*.

To ensure that PCs represent circular processes, we enforce a circular constraint, ($\mathbf{u}_{1i}^2 + \mathbf{u}_{2i}^2 = 1 \; \forall i \in \{1, \dots, N\}$), i.e., to constrain the samples in PC space to lie on a circle. We term the resulting $\mathbf{u}_1$ and $\mathbf{u}_2$ *cyclic principal components*. To identify rhythmic features, we constrain the loading vectors $\mathbf{v}_1$ and $\mathbf{v}_2$ to be suitably sparse, so that non-zero loading coefficients correspond to rhythmic features. Sparsity is enforced via an $l_1$ constraint with a hyperparameter $s$ to keep the optimization tractable – the smaller the value of $s$, the sparser the solution. The resulting optimization, which we term sparse cyclic PCA (scPCA), also

includes a $l_2$-regularization on the loading vectors ($\|\mathbf{v}_1\|_2 = \|\mathbf{v}_2\|_2 = 1$) to make the problem well-defined:

$$\max_{\mathbf{u}_1,\mathbf{v}_1,\mathbf{u}_2,\mathbf{v}_2,d} \left\{ 2d\mathbf{u}_1^T\mathbf{X}\mathbf{v}_1 + 2d\mathbf{u}_2^T\mathbf{X}\mathbf{v}_2 - d^2N - \overbrace{2d^2\mathbf{u}_1^T\mathbf{u}_2\mathbf{v}_1^T\mathbf{v}_2}^{\text{cross term}} \right\} \tag{2}$$

$$\text{s.t. } \|\mathbf{v}_1\|_2 = \|\mathbf{v}_2\|_2 = 1, \|\mathbf{v}_1\|_1 \leq s, \|\mathbf{v}_2\|_1 \leq s, \mathbf{u}_{1i}^2 + \mathbf{u}_{2i}^2 = 1, \forall i \in \{1,\dots,N\}.$$

Sparse cyclic PCA can be also considered an approximation of the periodic signals in the data by their Fourier series (at the fundamental frequency) under Gaussian noise or by a sinusoidal (cosinor) signal with additive noise (section A in S1 Text).

The optimization in Eq (2) is hard to solve due to the cross term. The cross term can be eliminated by imposing an additional orthogonality constraint on the loading vectors **v**s or on the cyclic principal components **u**s. But, such an optimization is not tractable. We therefore simply drop the cross term:

$$\max_{\mathbf{u}_1,\mathbf{v}_1,\mathbf{u}_2,\mathbf{v}_2,d} \left\{ 2d\mathbf{u}_1^T\mathbf{X}\mathbf{v}_1 + 2d\mathbf{u}_2^T\mathbf{X}\mathbf{v}_2 - d^2N \right\}$$

$$\text{s.t. } \|\mathbf{v}_1\|_2 = \|\mathbf{v}_2\|_2 = 1, \|\mathbf{v}_1\|_1 \leq s, \|\mathbf{v}_2\|_1 \leq s, \tag{3}$$

$$\mathbf{u}_{1i}^2 + \mathbf{u}_{2i}^2 = 1, \forall i \in \{1,\dots,N\}.$$

This simplified optimization in Eq (3) is bi-convex (section B in S1 Text) and can be solved to find a local maximum using Alternate Convex Search (alternating maximization) [46].

**Alternating maximization.** We can maximize Eq (3) by sequentially and iteratively updating the **u**s, the **v**s, and $d$ until convergence, starting from a random initial choice of vectors. The bi-convexity ensures that the score is monotonically non-decreasing during this update. For a given $(\mathbf{v}_1, \mathbf{v}_2)$, we can solve for $(\mathbf{u}_1, \mathbf{u}_2)$ by means of Lagrange multipliers to incorporate the circular constraints:

$$\max_{\mathbf{u}_1,\mathbf{u}_2,\{\mu_i\}} \left\{ \mathbf{u}_1^T\mathbf{X}\mathbf{v}_1 + \mathbf{u}_2^T\mathbf{X}\mathbf{v}_2 + \sum_i \mu_i(\mathbf{u}_{1i}^2 + \mathbf{u}_{2i}^2 - 1) \right\}$$

resulting in the update

$$\mathbf{u}_{1i} = \frac{\mathbf{y}_{1i}}{\sqrt{\mathbf{y}_{1i}^2 + \mathbf{y}_{2i}^2}} \text{ and } \mathbf{u}_{2i} = \frac{\mathbf{y}_{2i}}{\sqrt{\mathbf{y}_{1i}^2 + \mathbf{y}_{2i}^2}}, \forall i \in \{1,\dots,N\},$$

$$\text{where } \mathbf{y}_1 = \mathbf{X}\mathbf{v}_1, \mathbf{y}_2 = \mathbf{X}\mathbf{v}_2. \tag{4}$$

The solutions to $\mathbf{v}_1$ and $\mathbf{v}_2$ given $(\mathbf{u}_1, \mathbf{u}_2)$ are decoupled and need to maximize the projections $\mathbf{u}_1^T\mathbf{X}\mathbf{v}_1$ and $\mathbf{u}_2^T\mathbf{X}\mathbf{v}_2$ independently. For a given sparsity parameter $s$, the optimal **v** is proportional to $\mathbf{X}^T\mathbf{u}$ but with its elements soft-thresholded and scaled to satisfy both the $l_1$ and $l_2$ constraints, similar to sparse PCA [47]. (Given a threshold $\delta > 0$ and input $x \in \mathbb{R}$, the soft-thresholding function outputs $S(x, \delta) = (x - \delta)_+ - (-x - \delta)_+$). Guillemot et al. [48] provide a simple algorithm to compute the soft-thresholding parameter $\delta$ for any vector given the $l_1$ constraint $s$ resulting in an update of the form:

$$\mathbf{v}_1 = \frac{S(\mathbf{X}^T\mathbf{u}_1, \delta(s))}{\|S(\mathbf{X}^T\mathbf{u}_1, \delta(s))\|_2} \text{ and } \mathbf{v}_2 = \frac{S(\mathbf{X}^T\mathbf{u}_2, \delta(s))}{\|S(\mathbf{X}^T\mathbf{u}_2, \delta(s))\|_2} \tag{5}$$

Finally, given $\mathbf{u}_1, \mathbf{u}_2, \mathbf{v}_1, \mathbf{v}_2$, from Eq (3) the update for $d$ can be easily computed as:

$$d = \frac{\mathbf{u}_1^T \mathbf{X} \mathbf{v}_1 + \mathbf{u}_2^T \mathbf{X} \mathbf{v}_2}{N}. \tag{6}$$

**Relationship between the simplified optimization and the matrix approximation.** We dropped the cross term in the original scPCA optimization Eq (2) for computational convenience, but there are additional reasons to justify the simplification. As has been observed for the closely related sparse PCA, sparsity independently imposed on the **v**s often leads naturally to approximately orthogonal solutions and drives the $\mathbf{v}_1^T \mathbf{v}_2$ within the cross term to very small values [47]. Furthermore, explicitly enforcing orthogonality on the **v**s often leads to less sparse solutions, which is undesirable. Practically too, we observe the effect of this simplification to be negligible, since the neglected cross term constitutes less than 0.1% of the matrix approximation error in runs on synthetic data (S1I Fig) and on cancer data, (S6 Table). Nevertheless, it is advisable to ensure that this matrix approximation error is small when applying scPCA to new datasets. In summary, the scPCA procedure in practice finds excellent solutions to the matrix approximation problem Eq (1).

**Recovering the time labels:** The time labels for each sample $n \in \{1, \ldots, N\}$ in the interval [0,1] are computed as $\frac{1}{2\pi} \arctan\left(\frac{u_{2n}}{u_{1n}}\right)$.

**Identifying rhythmic features:** The features $i$ with $\mathbf{v}_{1i} \neq 0$ or $\mathbf{v}_{2i} \neq 0$ are classified as rhythmic. This is motivated by the association between scPCA and cosinor analysis [25] noted in S1 Text, where $\mathbf{v}_{1i}$ and $\mathbf{v}_{2i}$ correspond to the coefficients for the cosine and sine terms, respectively. Thus, arrhythmic features are defined as those with $\mathbf{v}_{1i} = \mathbf{v}_{2i} = 0$. The identification of rhythmic features depends on the sparsity parameter $s$ in scPCA, which is chosen by means of cross-validation (CV), as described below. CV is equivalent to a model selection-based formulation of rhythmicity analysis [49,50]. Thus, the CV procedure robustly chooses between the constant model ($\mathbf{v}_{1i} = \mathbf{v}_{2i} = 0$) and a rhythm model (non-zero $\mathbf{v}_{1i}$ or $\mathbf{v}_{2i}$) for each feature.

## Implementation - Cyclic Ordering with Feature Extraction (COFE)

The scPCA algorithm described above needs to address some additional issues before it can be used for rhythm profiling without a time series.

1. We standardize the features to have zero-mean and unit-variance (z-score) to ensure that they are equally weighted with respect to the $l_1$ constraint for sparsity.
2. The solution to the optimization, found via alternating maximization, only guarantees a local maximum. We therefore solve scPCA from multiple (*restarts* number of times) initial conditions for each choice of hyperparameter $s$, where the $(\mathbf{u}_{1i}, \mathbf{u}_{2i})$s are randomly initialized to lie on the unit circle and the elements **v**s are initialized from sparsity-inducing Laplacian distributions and scaled to satisfy the necessary constraints. We then choose the solution with the smallest matrix approximation error Eq (1) rather than the one with the best simplified objective function Eq (3) to mitigate to an extent the effect of the neglected cross term.
3. How do we choose a level of sparsity $s$ for the regularized problem Eq (3)? Since scPCA is an unsupervised learning technique, we do not have access to the true time

labels to help us choose $s$. Hence, we implemented a form of CV for unsupervised learning [51].

To perform K-fold CV, $1/K$ elements of the data matrix $\mathbf{X}$ are deleted in each fold. For the deleted elements to be evenly distributed across rows (samples) and columns (features), random deletion patterns are generated as circular row and column shifts of different pseudo-diagonals of the data matrix as defined in [52].

Deleted elements are initially imputed by column means (i.e, the feature means). We then alternately run scPCA and update the deleted elements using the scPCA-based approximation until both converge (within user-defined tolerances). The error between the original values of the deleted elements and their final imputed values from the low rank approximation serves as the CV error metric [51], which is a form of an expectation-maximization algorithm.

Repeated K-fold CV (*repeats* number of times) is performed for a range of $s$ values chosen by the user. The CV error (average imputation error) shows a sharp phase transition at the optimal $s$ (S1B Fig), when rhythms exist and all the rhythmic features above noise have been detected.

## Supporting information

**S1 Data. Data underlying individual panels of all Figures.**
(XLSX)

**S1 Text. Theoretical results.**
(PDF)

**S2 Text. Pseudocode of sparse cyclic PCA.**
(PDF)

**S3 Text. Performance evaluation methodology.**
(PDF)

**S4 Text. Analysis of TCGA data.**
(PDF)

**S1 Table. Time-series biological datasets used for benchmarking COFE.**
(PDF)

**S2 Table. Histological type of each AC included in this study.**
(PDF)

**S3 Table. Rhythmic genes identified by COFE, their circadian parameters and results of the enrichment analysis in each AC.**
(XLSX)

**S4 Table. Rhythmic proteins and protein forms important to cancer in each AC.**
(XLSX)

**S5 Table. FDA-approved, candidate and putative drugs along with their rhythmic targets in each AC.**
(XLSX)

**S6 Table. The neglected cross term in Eq (2) at optimal sparsity for the different datasets.**
(PDF)

**S7 Table. List of software packages used in COFE and subsequent analyses.**
(PDF)

**S1 Fig. COFE benchmarking on synthetic data.** (A–E) A typical COFE run on an example data set with $N$ = 2000 features, of which 400 were rhythmic. (A) Raw data for a few rhythmic features. (B) The output of repeated 5-fold cross-validation. (C) The underlying circular manifold reconstructed by COFE. COFE outputs the reconstructed phase within a cycle for each sample (D) and identifies rhythmic features in the data (E). (F) Temporal ordering performance of COFE measured using median absolute position error (MAPE) on data with $p$ = 2000 features and different signal-to-noise ratios (SNR), number of samples ($N$) and fraction of rhythmic features ($r$). Precision (G) and recall (H) performance for rhythmic feature identification for the same synthetic data in (F). (I) The ratio of the cross term dropped from Eq (2) as a fraction of the matrix approximation error for the synthetic data in (F). The data underlying Figure panels can be found in S1 Data.
(PDF)

**S2 Fig. Cross-validation (CV) quantifies the quality of the ordering.** The minimum imputation error (A) and optimal sparsity parameter (B) during CV for different combinations of signal-to-noise ratios (SNR), number of samples ($N$) and fraction of rhythmic features ($r$) in S1F Fig. (C) The relationship between the optimal sparsity parameter and the number of identified rhythmic features. (D) Output of repeated-5-fold CV for a synthetic dataset with only arrhythmic features (2000) for different number of samples ($N$). (E-H) Output of repeated-5-fold CV on datasets with 200 samples that were restricted to different fractions of the 24h cycle (see inset), and 2000 features, of which 400 were rhythmic. The data underlying Figure panels can be found in S1 Data.
(PDF)

**S3 Fig. COFE benchmarking against other methods.** (A) Temporal ordering performance of COFE compared to three other methods (CYCLOPS [15], PAGA [24] and scPrisma[21]) measured using median absolute position error (MAPE) on data with $p$ = 2000 features and different signal-to-noise ratios (SNR), number of samples ($N$) and fraction of rhythmic features ($r$). (B) Rhythmic feature detection performance of COFE compared to three other methods quantified by precision, recall and Matthews Correlation Coefficient (MCC), which combines precision and recall, for the same synthetic data in (A). The data underlying Figure panels can be found in S1 Data.
(PDF)

**S4 Fig. COFE benchmarking on biological data.** (A–C) COFE applied to the mouse liver RNA-seq data (SRP197108) collected hourly over 48h [53]. (A) The cyclic principal components (CPCs) extracted by COFE with the samples color-coded by phase within a cycle (normalized to 1.0). (B) The CPCs plotted against the true sample phase within a cycle (normalized to 1.0). (C) Scatter-plot of the true and predicted sample times. (D, E) Performance of COFE on human data. (D) COFE trained on Nanostring gene expression data from human blood monocytes is used to predict sample times for independent validation data [23]. (E) COFE applied to longitudinal time series human microarray gene expression data (GSE205155) from two different skin layers [13]. (F) Time label reconstruction of RNA-seq time-series of in-vitro cultures of four strains of *P. falciparum* (malaria parasite) [26]. The data underlying Figure panels can be found in S1 Data.
(PDF)

**S5 Fig. Rhythmic transcriptome in human adenocarcinomas (ACs).** Heatmaps of gene expression patterns of rhythmic genes, which are sorted by peak time of expression in each AC, for different ACs.
(PDF)

**S6 Fig. Circadian population rhythms in human adenocarcinomas (ACs).** (A) Distribution of peak phase of population rhythmic genes in different ACs. (B) The number of rhythmic genes with at least two-fold peak-to-trough amplitude in the ACs (compare with Fig 1C). (C) Raw data ordered using predicted time labels with the LOESS-smoothed estimates of the mean profile for selected genes. (D) Output of the repeated 5-fold cross-validation for each AC. The data underlying Figure panels can be found in S1 Data.
(PDF)

**S7 Fig. Coupling of circadian clock to the cell cycle and the proteome.** (A) Distribution of predicted time labels for the samples in each AC according to source country of the sample. (B) Phase difference predicted by COFE between cancer and patient-matched cancer samples not used for training COFE. (C) Predicted predominant cell cycle phases in the different patient samples predicted by the gene expression-based cell cycle phase scores. (D) Circular histograms of the peak time of expression of rhythmic proteins. The data underlying Figure panels can be found in S1 Data.
(PDF)

**S8 Fig. Rhythmic targets of putative cancer drugs.** The $\log_2$-fold amplitude (A) and peak time relative to the core clock (B) of the largest rhythmic target gene in each AC of putative drugs that target cancer pathways or processes. All putative drugs with rhythmic targets in at least 9 ACs are included. Drugs with multiple rhythmic gene targets in an AC are boxed in black. The data underlying Figure panels can be found in S1 Data.
(PDF)

## Author contributions

**Conceptualization:** Bharath Ananthasubramaniam, Ramji Venkataramanan.

**Data curation:** Bharath Ananthasubramaniam.

**Formal analysis:** Bharath Ananthasubramaniam.

**Funding acquisition:** Bharath Ananthasubramaniam.

**Investigation:** Bharath Ananthasubramaniam.

**Methodology:** Bharath Ananthasubramaniam, Ramji Venkataramanan.

**Software:** Bharath Ananthasubramaniam.

**Visualization:** Bharath Ananthasubramaniam.

**Writing – original draft:** Bharath Ananthasubramaniam.

**Writing – review & editing:** Ramji Venkataramanan.

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
