## [Editor Report · Decision Letter 0]

16 Jul 2024

Dear Dr Ananthasubramaniam, 

Thank you for submitting your manuscript entitled "Time series-free rhythm profiling using COFE reveals multi-omic circadian rhythms in in-vivo human cancers" for consideration as a Short Report by PLOS Biology.

Your manuscript has now been evaluated by the PLOS Biology editorial staff as well as by an academic editor with relevant expertise and I am writing to let you know that we would like to send your submission out for external peer review.

Once your full submission is complete, your paper will undergo a series of checks in preparation for peer review. After your manuscript has passed the checks it will be sent out for review. To provide the metadata for your submission, please Login to Editorial Manager (https://www.editorialmanager.com/pbiology) within two working days, i.e. by Jul 18 2024 11:59PM.

Kind regards,

Ines

--

Ines Alvarez-Garcia, PhD

Senior Editor

PLOS Biology

---

## [Decision Letter · Decision Letter 1]

19 Sep 2024

Dear Dr Ananthasubramaniam,

Thank you for your patience while your manuscript entitled "Time series-free rhythm profiling using COFE reveals multi-omic circadian rhythms in in-vivo human cancers" was peer-reviewed at PLOS Biology. Please also accept our appologies for the delay in providing you with our decision. The manuscript has now been evaluated by the PLOS Biology editors, an Academic Editor with relevant expertise, and by three independent reviewers. 

The reviews are attached below. As you will see, the reviewers find the method novel and interesting, but they also raise several issues that would need to be addressed before we can consider the manuscript for publication. Reviewers 1 and 2 would like to see comparisons with additional methods published in terms of novelty and advance, and Reviewer 1 also thinks that the claims should be experimentally validated and confirm that the method makes valid predictions. In addition, Reviewer 2 asks for further details on the method to allow readers to reproduce it, and suggests some improvements that would be important to clarify the results. Reviewer 3 thinks you should explore the performance of COFE as a function of the observed fraction of cycle in simulated data, along with discussing the implications for the TCGA application and clarifying several points.

In light of the reviews, we would like to invite you to revise the work to thoroughly address the reviewers' reports. Given the extent of revision needed, we cannot make a decision about publication until we have seen the revised manuscript and your response to the reviewers' comments. Your revised manuscript is likely to be sent for further evaluation by all or a subset of the reviewers.

**IMPORTANT - SUBMITTING YOUR REVISION**

3. Resubmission Checklist

a) *PLOS Data Policy*

b) *Published Peer Review*

d) *Blurb*

Please also provide a blurb which (if accepted) will be included in our weekly and monthly Electronic Table of Contents, sent out to readers of PLOS Biology, and may be used to promote your article in social media. The blurb should be about 30-40 words long and is subject to editorial changes. It should, without exaggeration, entice people to read your manuscript. It should not be redundant with the title and should not contain acronyms or abbreviations. For examples, view our author guidelines: https://journals.plos.org/plosbiology/s/revising-your-manuscript#loc-blurb

Sincerely,

Ines

--

Ines Alvarez-Garcia, PhD

Senior Editor

PLOS Biology

Reviewers' comments

Rev. 1:

This manuscript presents a method, COFE, for time-series free circadian analysis particularly of omics data. The method relies on determining two principle components, and in that way it is similar in spirit to other methods. The authors then apply the method to existing databases, such as the TCGA or Reactome database. From this, they make claims about cyclic components within these databases and, for example, the relationship between cycling in the cell cycle and circadian clock.

I struggled quite a bit with this manuscript. While on the surface, there are claims that might be of broad interest to the readership of PLOS Biology, few in the time series or signal processing communities would accept their method as is. Indeed, there are large communities devoted to this kind of work and questions of inferring rhythms from limited data, the SAMPTA conference might be the best, and there are many mathematical concerns that are not addressed such as topological considerations, questions about the shape of rhythms and harmonics, and the relationship to other methods. There is some benchmarking against other methods. It is not always completely clear that this is a better method, and there are many additional methods, even especially for circadian analysis, to compare to. The study of synthetic data needs to be significantly expanded to be confident in the method. I'd be less concerned about this if the manuscript was focused on a single biological question and there was other new data to support the predictions of the method.

No new data is presented, and even if the method were more mathematically validated as others are, I think PLOS Biology readers would want some further experimental validation of their claims. One promising feature is that the circadian genes have phases similar to those found in other studies, but this is not enough for the very broad claims that are made in Figure 3 about candidate cancer drugs.

In summary, a lot more work needs to be done before it can be a convincing method broadly used. I encourage the authors to work towards this. I suggest breaking off the parts of the manuscript, focusing on the method, and getting validation by publishing them in the time series literature or statistics literature as a general method. I would then consider a manuscript where more biological details can be explored (perhaps in a full manuscript format), for example, through new data or strong supporting evidence from other studies. With that, readers would see that the method makes validated predictions, other readers would be interested in using the methods in other contexts, and such a manuscript might find a welcome home in PLOS Biology or a similar journal.

Rev. 2:

Single-cell RNA sequencing (scRNA-seq) data typically provides a snapshot of cellular states without temporal information. However, given its massive amount of information, recent efforts have focused on extracting temporal dynamics, such as circadian rhythms, from scRNA-seq data. The author introduced COFE (Cyclic Ordering with Feature Extraction), which orders cells according to their circadian phase while simultaneously identifying rhythmic features. Applying COFE to 11 different human adenocarcinomas (ACs) identified 2,000 to 6,000 rhythmic genes in each AC. The amplitude and peak time of these rhythmic genes were further analyzed using Gene Set Enrichment Analysis (GSEA) with the Reactome database, leading to the identification of enriched pathways associated with metabolism, the cell cycle, and the immune system. The results are very interesting and the developed method is very useful. However, the explanation for the method seems not sufficient to understand the algorithm fully and reproduce the results.

Major:

1. The author developed COFE for pseudo-time inference in scRNA-seq data. Although a benchmark study was performed comparing this algorithm with CYCLOPS, there are numerous other popular algorithms for pseudo-time inference, ranging from conventional trajectory analysis methods such as PAGA [1] to newer approaches such as scPrisma [2]. Therefore, performing an additional benchmarking study that includes widely used tools, such as PAGA and scPrisma, would help to better demonstrate the novelty of COFE.

2. page 12 - 3rd paragraph (2nd line) In "Theory" section of the Materials and Methods: The authors ignored the term related to the orthogonality of singular vectors in equation (1) simply because they are difficult to deal with. This omission could compromise optimal data compression, as well as amplify periodic signals by exaggerating certain principal components (PCs). Therefore, please explain in more detail the reason for the omission of these terms or cite the paper if possible.

3. page 13 - 1st paragraph (1st line) In "Alternating maximization" section of the Materials and Methods: There is no explanation of which distribution was used to generate the initial conditions for u and v. Furthermore, the following steps are difficult to understand based on the current description, particularly regarding the algorithm used for maximizing equation (2). For reproducibility, it would be helpful to provide a detailed explanation of how the u and v vectors are initialized and updated, as well as which algorithm was used for the maximization, possibly including pseudo-code or a flow chart.

4. page 13 - 1st paragraph (1st line) In "Alternating maximization with regularization" section: If v is not a binary vector, it seems too arbitrary to select the gene corresponding to a non-zero element of v as a rhythmic gene. For instance, if an element of v is close to zero but not exactly zero (e.g., 0.0000001), can it truly be considered a rhythmic gene?

5. page 14 - 1st line in (iv) of "Implementation - Cyclic Ordering with Feature Extraction (COFE)" section: The author introduces s as the sparsity level. However, as s decreases, the l1 norm of vector v also decreases, which, according to equation (6), indicates a higher level of sparsity. To clarify this, it would be beneficial to elaborate on the relationship between the parameter s and the level of data sparsity in the manuscript.

6. page 14 - 3rd line in (iv) of "Implementation—Cyclic Ordering with Feature Extraction (COFE)" section: There is no description of how random elements were deleted from the data for cross-validation (CV). How were these elements selected and deleted? Were elements deleted regardless of their value, or did the probability of deletion vary based on the magnitude of the element's value? Additionally, is there a relationship between the parameter s and the number of deleted elements?

7. page 3 and 4 - in Widespread in vivo population rhythms in human adenocarcinomas rhythmic gene section: After applying COFE, the author extracted the amplitudes and peak times from genes ordered by their pseudo-time and used these quantities for analysis. However, there is no explanation of how the amplitude and peak time of each gene were determined. As these quantities are used extensively in this study, it is essential to describe the steps for obtaining amplitude and peak time from the ordered genes.

Minor:

1. page 4: Figure 1A does not clearly explain the method. A dedicated figure illustrating the different steps of the algorithm would likely be more effective. For example, the v vectors were incorrectly described as binary, but they are not in Fig. 1a. Additionally, the description of the grey panel used to determine the sparsity parameter in Fig. 1A is missing from both the caption and the result section, making it difficult to interpret the role of this step in the COFE. It would be better to add this description to the figure caption.

2. page 4: The vertical colored line in Figure 1F makes it difficult to determine which data has been selected.

3. page 3 - 1st paragraph (4th line) in "Widespread in vivo population rhythms in human adenocarcinomas" section: « which are cancers of epithelial origin that develop in organs and glands and account for 80 to 90 percent of all cancers »: reference?

4. page 6 - 1st paragraph (1st line) in "Key cellular processes are enriched for rhythmic genes" section: « To identify the processes enriched for rhythmic genes in different ACs, we performed gene set enrichment analysis [25] using the Reactome database »

a. Put the database version as it is frequently updated

b. What tools and GMT files are used? (Java application, R package, ...)

5. 13 page - 1st line in (i) of "Implementation - Cyclic Ordering with Feature Extraction (COFE)" section: Before applying COFE, data was standardized. However, the specific standardization method used is not clearly described, even though various approaches, such as log-normalization and scTransform, are available to stabilize data variance. Furthermore, standardization can distort signals in sparse data, potentially leading to false discovery, as noted in a recent study [3]. Therefore, it is necessary to specify the method used for data standardization and to explain how any potential distortion in sparse data was addressed.

6. If available, put the version of recount3

7. page 8 - 2 paragraph of "Protin expression is rhythmic, even in ACs with few rhythmic genes": The authors wrote, "Activated forms of members of key signaling pathways, such as AKT, EGFR, and MAPK, were also rhythmic in almost all ACs." These three pathways are well-known to be deregulated in cancer. More information about the different proteins involved in these pathways and found rhythmic could be added. The authors may want to add the name of the most important rhythmic proteins, the results they found, their involvement, and the consequences on the pathway. They can further develop by showing how this is relevant for cancer studies.

References:

1. Wolf, F.A., Hamey, F.K., Plass, M. et al. PAGA: graph abstraction reconciles clustering with trajectory inference through a topology preserving map of single cells. Genome Biol 20, 59 (2019). https://doi.org/10.1186/s13059-019-1663-x

2. Karin, J., Bornfeld, Y. & Nitzan, M. scPrisma infers, filters and enhances topological signals in single-cell data using spectral template matching. Nat Biotechnol 41, 1645-1654 (2023). https://doi.org/10.1038/s41587-023-01663-5

3. Kim, H., Chang, W., Chae, S.J. et al. scLENS: data-driven signal detection for unbiased scRNA-seq data analysis. Nat Commun 15, 3575 (2024). https://doi.org/10.1038/s41467-024-47884-3

Rev. 3:

This paper presents a novel method for reconstructing temporal information in a population of untimed high-dimensional (eg, RNAseq) data by projecting it onto a circular manifold. By enforcing sparsity in the projection, it is possible to recover the features that are the most strongly periodic, without the need for a second stage analysis. Applied to data from TCGA, the authors discover rhythmicity in tumor cell gene expression that differs between healthy and normal tissue; because the proteins associated with this rhythmic expression may be chemotherapy targets, this points the way toward timed dosing regimens that may be more effective with fewer side-effects. All in all, I find this to be an excellent contribution to the field.

I do have a few questions on the methodological side:

* The projection of the samples onto the circle assumes that they are collected "around the clock". What happens if they are not, that is, if only a portion of the cycle is observed? With the possible exception of postmortem samples, the vast majority of samples will tend to be collected between 9am and 5pm -- only 1/3 of the cycle observed. The performance of COFE as a function of the observed fraction of cycle should be explored in simulated data, and the implications for the TCGA application should be discussed.

* COFE's performance was investigated in synthetic data with as little as r=0.1 fraction of cycling features. What happens if there are NO cycling features, such as in a case where (for example) the circadian rhythm is severely compromised? It will of course map the data onto a circle (because it has no choice but to do so), but then what? And can one tell if there is NOT a strong signal? (Quantification of the amount of variation explained by the two circular PCs may be useful.)

* How does COFE perform if there are multiple rhythms with different frequencies present in the data?

* The optimization finds only local minima, and thus depends on the initialization; the authors report that they perform multiple restarts and "pick the solution with the best score," which I assume means the one with the greatest maximum of the objective function. How many restarts, and how rugged is the landscape?

---

## [Decision Letter · Decision Letter 2]

2 Apr 2025

Dear Dr Ananthasubramaniam,

Thank you for your patience while we considered your revised manuscript entitled "Time series-free rhythm profiling using COFE reveals multi-omic circadian rhythms in in vivo human cancers" for publication as a Short Report at PLOS Biology. Please also accept my apologies for the long delay in sending you our decision. This revised version of your manuscript has been evaluated by the PLOS Biology editors, the Academic Editor and the original reviewers.

Based on the reviews and follow up discussions with the Academic Editor, we are likely to accept this manuscript for publication, provided you address the remaining points raised by Reviewer 2. While we do appreciate the points raised by Reviewer 1, we do think the scope is appropriate for the journal. Please also make sure to address also the data and other policy-related requests stated below my signature.

In addition, we would like you to consider a suggestion to improve the title:

"Rhythm profiling using COFE reveals multi-omic circadian rhythms in human cancers in vivo"

We expect to receive your revised manuscript within two weeks.

*Published Peer Review History*

*Press*

Sincerely,

Ines

--

Ines Alvarez-Garcia, PhD

Senior Editor

PLOS Biology

Fig. 2B-E, G; Fig. 32A-G; Fig. 4A, B; Fig. S1A-I; Fig. S2A-H; Fig. S3A, B; Fig. S4A-F; Fig. S5; Fig. S6A-D; Fig. S7A-D and Fig. S8A, B

**Please also obtain a doi in Zenodo for data or software deposited in Github by following these instructions:

https://cassgvp.github.io/github-for-collaborative-documentation/docs/tut/6-Zenodo-integration.html

Reviewers' comments

Rev. 1:

I still struggle with this manuscript. First about novelty. Just to recap, some methods already estimate the circadian phase from a single sample, and other methods for pseudo-time inference, so while there is some novelty here it isn't clear to me quite how much, especially with methods being based at least partially on cosinor analysis in the end. The authors did add some comparison (e.g., panel S6D), but in my eyes, this shows, at best, that it is comparable to other methods. 

The next question (perhaps the biggest) I struggled with is journal fit: At least as I understand it, PLOS Computational Biology is a great journal that publishes rigorous methods to analyze biological data. It seeks methods that can have a large impact on biology, and where the details of the computational methods can be fully explored. For example, there is a methods section where "Methods articles present outstanding contributions of innovative computational methods for highly relevant biological problems." "The methods are expected to be thoroughly validated on real (and, in addition, possibly artificial) data and to be generally available for download or as web services."

PLOS Biology seeks biological insight, where, the readers would want more than generating a method and applying it to a publically available dataset. Some paradigm changing hypothesis that could cause us to rethink a biological system, new experimental data, etc. The editor correctly summed up my concern by saying, ""Reviewer 1 also thinks that the claims should be experimentally validated and confirm that the method makes valid predictions." As I mentioned, "With that, readers would see that the method makes validated predictions" So, I thought if they could show that one of their predictions was correct and lead to follow work or some new advance (rather than just a broad claim), it might excite the readers of PLOS Biology. 

Unfortunately, the authors didn't follow this up. In my mind, there are many assumptions in the mathematical method and their analysis. Gene set enrichment is also far from perfect. The claims they make are broad. To me, they are not nearly strong enough for an experimental group to risk spending significant time or limited funds on following up. It doesn't seem to meet the bar to be "novel, provocative and of general interest, in such a way as to spur future research"

Moreover they seem to argue that any validation would be specific to human and specific tissues, which in my mind only limits their method. For example, other single-sample methods to assess circadian phase have received attention because they might be able to be widely used as circadian markers. But it seems the authors have a much more limited scope. They do mention that some other papers they compare themselves to did not do as much mathematical validation, for example, PAGA, but these unlocked large datasets for which there was a large demand, whereas they seem focused on a question of limited scope.

I wrote "many mathematical concerns that are not addressed such as topological considerations, questions about the shape of rhythms and harmonics, and the relationship to other methods." and again the question is whether these should be addressed here or at another journal. For example, I suggested SAMPTA (perhaps PLOS Computational Biology might also be good). They responded, "We also perused the proceedings of SAMPTA from 2013 onwards, and were unable to find methods that can be applied for inferring rhythms in data without time labels." Again, I wouldn't expect their method to have been previously published in SAMPTA, but there the do get into the many details of timeseries methods, would be a more appropriate place to check mathematical methods and would be very wary of unrigorous statements the authors made in response, for example like "negligible effect of the approximations we made in the formulation of COFE."

Indeed, I think this could be a good contribution to another journal. I write this since I think if it were in the right journal it could find a good home and audience, and if they got this right the manuscript would have more impact, citations etc. The editors would know best, and I'd go with their judgement in case they see things different or if I am missing something.

Rev. 2:

We thank the authors for their detailed responses and revisions to the manuscript based on the reviewer comments. Overall, the manuscript is significantly improved. 

However, we would like to offer further comments on the responses to Reviewer 2's Major Comments #2 and #4:

1. Regarding the response to Major Comment #2: We acknowledge the authors' thorough explanation justifying the simplification of the optimization objective (Eq. 2 to Eq. 3) by dropping the cross-term related to orthogonality. The justification, based on the nature of sparse PCA where sparsity constraints often lead to near-orthogonality (v_1^T v_2≈0), the empirical evidence showing the negligible contribution (<0.1%) of the term to the overall matrix approximation error, and the best solution selection, seems sufficient within the practical context of achieving sparsity via scPCA. 

However, upon examining Figure S1I, we note a potential trend where the fractional contribution of the neglected term appears to increase slightly with higher SNR. While the absolute magnitude remains very small, as stated by the authors, this relative trend warrants a brief mention. Thus, we suggest adding a brief cautionary note in the Methods or Discussion section to acknowledge this observation.

2. Regarding the response to Major Comment #4: In response to Major Comment #4, the authors justified not adding an arbitrary threshold beyond the l_1 constraint itself by arguing (a) that the constraint inherently performs thresholding consistent with standard sparse PCA practice, and (b) that their cross-validation results show the optimal sparsity level is data-driven, with further thresholding increasing error and thus being suboptimal.

However, our original question was not about suggesting that a different threshold for rhythmicity should be set. The comment extends beyond simply "how to determine the threshold" and poses the more fundamental question: "What is the justification for considering a non-zero value in the loading vector (v) as sufficient evidence for rhythmicity?" Therefore, an explanation is required as to how the mathematical condition v_j=0 can represent the biological (or statistical) phenomenon of 'rhythmicity' within the COFE model framework itself. We request the authors to elaborate on this fundamental link.

We believe addressing these two points will further enhance the manuscript's clarity and rigor. We commend the authors again for addressing the other comments effectively.

---

## [Editor Report · Decision Letter 3]

3 May 2025

Dear Dr Ananthasubramaniam,

Thank you for the submission of your revised Short Report entitled "Rhythm profiling using COFE reveals multi-omic circadian rhythms in human cancers in vivo" for publication in PLOS Biology. On behalf of my colleagues and the Academic Editor, Martha Merrow, I am delighted to let you know that we can in principle accept your manuscript for publication, provided you address any remaining formatting and reporting issues. These will be detailed in an email you should receive within 2-3 business days from our colleagues in the journal operations team; no action is required from you until then. Please note that we will not be able to formally accept your manuscript and schedule it for publication until you have completed any requested changes.

PRESS

Sincerely, 

Ines

--

Ines Alvarez-Garcia, PhD

Senior Editor

PLOS Biology
